# Outdoor Vision-and-Language Navigation Needs Object-Level Alignment

**DOI:** 10.3390/s23136028

**Published:** 2023-06-29

**Authors:** Yanjun Sun, Yue Qiu, Yoshimitsu Aoki, Hirokatsu Kataoka

**Affiliations:** 1Department of Electronics and Electrical Engineering, Faculty of Science and Technology, Keio University, 3-14-1, Hiyoshi, Kohoku-ku, Yokohama 223-8522, Japan; aoki@elec.keio.ac.jp; 2National Institute of Advanced Industrial Science and Technology (AIST), Tsukuba 305-8560, Japan; qiu.yue@aist.go.jp (Y.Q.); hirokatsu.kataoka@aist.go.jp (H.K.)

**Keywords:** vision-and-language navigation, landmark-based navigation, embodied AI

## Abstract

In the field of embodied AI, vision-and-language navigation (VLN) is a crucial and challenging multi-modal task. Specifically, outdoor VLN involves an agent navigating within a graph-based environment, while simultaneously interpreting information from real-world urban environments and natural language instructions. Existing outdoor VLN models predict actions using a combination of panorama and instruction features. However, these methods may cause the agent to struggle to understand complicated outdoor environments and ignore the details in the environments to fail to navigate. Human navigation often involves the use of specific objects as reference landmarks when navigating to unfamiliar places, providing a more rational and efficient approach to navigation. Inspired by this natural human behavior, we propose an object-level alignment module (OAlM), which guides the agent to focus more on object tokens mentioned in the instructions and recognize these landmarks during navigation. By treating these landmarks as sub-goals, our method effectively decomposes a long-range path into a series of shorter paths, ultimately improving the agent’s overall performance. In addition to enabling better object recognition and alignment, our proposed OAlM also fosters a more robust and adaptable agent capable of navigating complex environments. This adaptability is particularly crucial for real-world applications where environmental conditions can be unpredictable and varied. Experimental results show our OAlM is a more object-focused model, and our approach outperforms all metrics on a challenging outdoor VLN Touchdown dataset, exceeding the baseline by 3.19% on task completion (TC). These results highlight the potential of leveraging object-level information in the form of sub-goals to improve navigation performance in embodied AI systems, paving the way for more advanced and efficient outdoor navigation.

## 1. Introduction

Vision-and-language navigation (VLN) is a challenging task wherein agents must navigate a real-world visual environment by following instructions in natural language. The agent must comprehend the instructions and ground them in the observable environment with only visual perception. Moreover, it needs to reason about its position in relation to objects and understand how these relationships change as it moves through the environment. Ultimately, the agent should generate a series of corrective actions to reach the destination. Enabling agents to understand the relationship between instructional words and perceived features in the environment is one of the critical challenges of VLN [1].

To help the agent understand the relationship between instructions and environmental features, approaches such as the OAAM (object-and-action aware model) [2] and ROAA (room-and-object aware attention) [3] extract the semantic connections between different modalities (actions, scenes, observed objects, and objects mentioned in instructions) and then softly align them with an attention mechanism. However, their effectiveness is limited in outdoor environments due to inherent differences in environmental conditions. Compared to indoor VLN [4], outdoor VLN faces a more complex real-world urban environment, and it is considered a more challenging task for an agent to be able to handle more spatial information and significant visual feature differences such as textures, lighting, and scene layouts [5,6]. Furthermore, outdoor environments also involve dynamic elements such as pedestrians and vehicles, which indoor models may struggle to handle effectively.

As illustrated in Figure 1 [6,7,8,9,10], which largely relies on simplistic encoder–decoder models, our approach places significant emphasis on landmarks. These existing models concatenate instruction features and panoramic features and use a decoder to predict a sequence of actions from these concatenated features. While these models can produce agents capable of basic navigation, their capacity to understand specific semantics is lacking. However, such an approach does not align with human intuition. When navigating to a place in an unfamiliar environment, humans tend to use visible objects such as buildings and other objects seen along the way as reference landmarks [11]. Therefore, the objects are essential clues in outdoor VLN. We observed that existing models hardly focus on object tokens by visualizing the paths generated by them and the attention heatmap of the instructions. In many cases, the agent ignores landmarks given by the instructions, making turns or stopping in incorrect places.

Inspired by humans’ use of landmarks to navigate unfamiliar places and the significant results of object-aware indoor VLN models [2,3,12,13], several recent studies of indoor VLN models in particular [14,15,16] have demonstrated the utility of using object detectors. Therefore, we believe that better use of landmarks in outdoor VLN scenarios is a more rational way to navigate, and may improve navigation performance.

We, therefore, proposed an **O**bject-level **Al**ignment **M**odule (OAlM), which is a straightforward yet effective module that guides the agent to focus more on the landmarks indicated in the navigation instructions. The OAlM consists of four components: the landmark extractor, landmark detector, match checker, and index controller. The landmark extractor identifies and extracts landmarks from the instructions, which serve as critical reference points that the agent should aim to pass during navigation. Simultaneously, a landmark detector scans the environment and identifies landmarks, such as text on signboards or objects explicitly mentioned in the instructions. The index controller indicates which reference landmark the agent should aim for next. It effectively sequences the order of landmarks to guide the navigation process. Finally, the match checker is responsible for assessing whether the agent has successfully reached the current reference landmark at each step. Based on this verification, the index controller adjusts the points accordingly.

We have extensively experimented on the Touchdown dataset [6] with three baseline outdoor VLN models, RCONCAT [6], GA [7], and ORAR [10]. We added the OAlM module into the baselines and compared the results with/without our OAlM to show that landmark alignment is necessary for better outdoor VLN. Our ablation studies and qualitative results further verify that the improvement indeed comes from agents being able to make better use of landmarks. In summary, the main contributions of this paper are as follows:We developed an object-level alignment module (OAlM) and used OCR and object-detection technology to identify the prominent landmarks. OAlM pays more attention to landmarks and helps the agent understand the environment to improve navigation ability.We extensively evaluated our approach and found that it outperforms all metrics on the Touchdown dataset against multiple existing methods, exceeding the baseline by 3.19% on task completion (TC).

## 2. Related Works

### 2.1. Vision-and-Language Navigation

An agent needs to reach the target destination by following a set of natural language instructions. One of the foundational benchmarks in this area, Room-to-Room (R2R) [4] for indoor scenarios was proposed, along with a multimodal Seq2Seq baseline model. Then several indoor VLN datasets were created by extending R2R with other languages, such as XL-R2R [17] and RxR [18]. However, outdoor environments are usually more complex and diverse compared to indoor environments. Recognizing this, the Touchdown [6] dataset was proposed, making the first outdoor VLN benchmark, which is based on Google Street View (https://developers.google.com/maps/documentation/streetview/intro, accessed on 10 September 2022). Subsequently, additional outdoor VLN datasets, such as StreetLearn [5,19], StreetNav [20], Map2seq [21], and Talk2Nav [22] for outdoor VLN have also been proposed.

### 2.2. Outdoor Vision-and-Language Navigation Models

Numerous methods have been proposed to address the outdoor VLN task. RCONCAT [6] is the baseline model proposed in the original Touchdown paper, and it encodes the trajectory and the instructions in an LSTM-based model. ARC [8] proposed ARC+l2s, which cascades the action prediction into a binary stopping decision and subsequent direction classification. VLN-Transformer [9] uses a pretrained BERT [23] applied to an external multimodal dataset to enrich the navigation data. GA [7] uses gated attention to compute a fused representation of instructions and images to predict actions. ORAR [10] adds junction-type embedding and a heading delta to train a general model, reducing the performance gap between seen and unseen environments. However, these outdoor VLN models use LSTM or other encoders to encode the instructions and a decoder LSTM to predict the actions. This architecture limits attention to particular semantics, such as objects, which are essential clues in VLN.

### 2.3. Object-Aware Vision-and-Language Navigation

To handle fine-grained information, such as objects, in order to facilitate simultaneous understanding of instructions and visuals, vision-and-language pretrained models such as ViLBERT [24] have been widely adopted in VLN because they can provide good joint features for better understanding of instructions and environmental features. ORIST [25] was trained with specific features such as objects and rooms. OAAM [2] extracted object and action tokens from instructions and then matched them to visual perceptions and orientation to predict actions. ROAA [3] was designed to perceive the room and object-type information from instruction and visual perceptions, while the knowledge-enabled entity relationship reasoning (KERR) module was designed to learn the internal–external correlations between room and object for the agent to take appropriate actions. SOAT [12] uses a scene classification network and an object detector to produce features that match these two distinct visual cues. M-Track [26] was proposed as a milestone builder and milestone checker to guide agents complete a task step by step and was tested on the ALFRED [27] dataset. The scenario involves an agent completing household tasks in an interactive visual environment. While these approaches all perform well in indoor scenarios, they are unsuitable for outdoor VLN, which has a different and more complex environmental graph. The vast urban areas being navigated lead to a much larger space for the agent to explore and also contain a wider range of encountered objects in the visual environment. Therefore, outdoor VLN requires special treatment in recognizing these diverse objects and using them to understand the environment better.

## 3. Proposed Methods: OAlM

In this section, we introduce the model that we use to analyze navigation performance in outdoor VLN. First, we give a definition for the VLN problem. Afterwards, we provide a more formal description of the model architecture.

### 3.1. VLN Problem Definition

The VLN task requires an agent in an environment to follow natural language instructions $X=\left\{x_{1},x_{2},\ldots ,x_{l}\right\}$, along a trajectory and stop at a target location. The length of the instructions is denoted by *l*, and a word token is denoted by $x_{i}$. The environment is an undirected graph consisting of nodes v∈V and labeled edges (v,u)∈E, where *u* is one of the outgoing neighbors of node *v*. Each node is associated with a panoramic RGB image and each edge connects nodes to neighboring panoramas at the heading angle $\alpha_{\left(v,u\right)}$. At time *t*, the agent’s state is defined by $s_{t}\in S,s_{t}=\left(v_{t},\alpha_{\left(v_{t-1},v_{t}\right)}\right)$, where $v_{t}$ is the agent-observed panoramic view, and $\alpha_{\left(v_{t-1},v_{t}\right)}$ is the heading angle between previous state’s view $v_{t-1}$ and the current state’s view $v_{t}$. Following the instructions *X*, the agent executes an action $a_{t}\in \left\{\mathrm{FORWARD},\;\mathrm{LEFT},\;\mathrm{RIGHT},\;\mathrm{STOP}\right\}$ and is updated to the next state $s_{t+1}$. The agent must produce a sequence of state-action pairs $\langle \left(s_{1},a_{1}\right),\left(s_{2},a_{2}\right),\ldots ,\left(s_{n},a_{n}\right)\rangle$, where $a_{n}=\mathrm{STOP}$, to reach the goal location.

### 3.2. Overview

As shown in Figure 2, the model takes navigation instructions as the input and outputs a sequence of agent actions. At each decoding timestep, a new visual representation of the current agent state within the environment is computed, and the alignment between the objects in the environment and the given instructions is computed. The action predictor then decodes a fused text and visual representation and eventually predicts the next action.

The intuition behind the architecture of this model is to first concatenate visual and instruction features at each time step because high-level features (e.g., objects) can outperform lower-level features extracted by the CNN [14]. We then use these high-level features to check whether the agent is on the correct route. According to the results of this check and the fused features, the Action-Predictor decides what action to take. We also propose an object-level alignment module (OAlM), which allows the model to pay more attention to object information from the environment and the instructions.

### 3.3. OAlM

The alignment module is made up of four parts: a landmark extractor, a landmark detector, a match checker, and an index controller. The landmark extractor and landmark detector detect landmarks from the instructions and the panoramas. We set the ordered landmarks mentioned in the instructions as references. At each timestep, the match checker determines whether the agent has reached the reference by comparing it to the observed landmark in the environment. The index controller sets the first landmark as the current reference landmark at the start of navigation, and updates to the next landmark when the agent reaches the current reference landmark.

**Landmark Detector:** At each timestep *t*, the landmark detector identifies the text on signboards and objects at the current node $v_{t}$ as ${\mathrm{LM}}_{v_{t}}$. To obtain the high quality ${\mathrm{LM}}_{v_{t}}$, we set a threshold.
(1)$${\mathrm{LM}}_{v_{t}}=\mathrm{Landmark-Detector}\left(v_{t}\right)$$
${\mathrm{LM}}_{v_{t}}=\left\{{\mathrm{lm}}_{v_{t}}^{\left(1\right)},{\mathrm{lm}}_{v_{t}}^{\left(2\right)},\cdots ,{\mathrm{lm}}_{v_{t}}^{\left(n\right)}\right\}$ is a set of observed landmarks, where *n* is the maximum number of landmarks at a panoramic view location $v_{t}$.

**Landmark Extractor:** To extract this sequence from the natural language instructions, we employed a standard large language model, which in our prototype was GPT-3 [28]. We used a prompt with three examples of correct landmark extractions, and the model was then used to extract a list of landmarks from the instructions. We evaluated the performance on 20 test pairs, the accuracy of GPT-3 outputs was 99%. Therefore, the model’s output was reliable and robust to small changes in the input prompts. For instance, if the prompts give the instructions *“Continue down the long block to the next light with Starbucks on the near left corner. Turn right, then stop. Dunkin’ Donuts should be ahead on your right”*, the landmark extractor will output *“Starbucks, Dunkin’ Donuts”* as the ordered landmarks. In this case, the agent would check to see if it has arrived at *Starbucks* first and then check *Dunkin’ Donuts*. We extract the list of ordered landmarks ${\mathrm{LM}}_{X}=\left\{{\mathrm{lm}}_{X}^{\left(1\right)},{\mathrm{lm}}_{X}^{\left(2\right)},\cdots ,{\mathrm{lm}}_{X}^{\left(d\right)}\right\}$ from instructions *X*, where *d* is the maximum number of ordered landmarks extracted.

**Match Checker and Index Controller:** The match checker determines if an agent has reached a known landmark, while the index controller keeps track of the reference landmark that the agent should arrive at next. At the beginning of navigation, the index controller points to the first of the ordered landmarks ${\mathrm{lm}}_{X}^{\left(1\right)}$ obtained using the landmark extractor.
(2)$$\begin{array}{cc} \; & \cos\left(A,B\right)=\frac{A\cdot B}{∥A∥∥B∥} \end{array}$$
(3)$$\begin{array}{cc} \; & m_{t}=\frac{1}{n}\sum_{i=1}^{n}\cos\left({\mathrm{lm}}_{v_{t}}^{\left(i\right)},{\mathrm{lm}}_{X}^{\left(t\right)}\right) \end{array}$$

With Equation (Equation 2), the match checker calculates the cosine similarity between all of the landmarks observed in the environment ${\mathrm{LM}}_{v_{t}}$ and ${\mathrm{lm}}_{X}^{\left(t\right)}$ and uses this as a matching score $m_{t}$ to check if the agent has reached the current reference landmark ${\mathrm{lm}}_{X}^{\left(t\right)}$. If the matching score $m_{t}$ is over the set threshold, the index controller will indicate to the agent the next reference landmark ${\mathrm{lm}}_{X}^{\left(t+1\right)},t<d-1$ until the agent reaches the last landmark.

### 3.4. Inference of OAlM

We implemented our OAlM for use with the following existing VLN models: GA [7], RCONCAT [6], and ORAR [10]. These models were chosen because of their similar architectures and allow seamless integration of our OAlM. To explain in detail, these models embed and encode the instructions *X* by LSTM [29] or Bi-LSTM [30] to obtain instruction features $X_{t}$. Following each timestep *t*, these models utilize a pretrained ResNet-50 [31] or CNNs to extract panorama features $v_{t}$. Subsequently, the instruction features $X_{t}$, and the panorama features $v_{t}$ are concatenated to predict the action $a_{t}$ at each timestep *t*, as indicated by Equation (Equation 4). In this euqation, the Action-Predictor is represented as a simple FFNN layer.

In our implementation, we add the matching score $m_{t}$, introduced above. The Match Checker calculates the matching score and helps the model to confirm whether the sub-goal has been reached. We concatenate $m_{t}$ with the panorama features $v_{t}$, and the instruction features $X_{t}$, as expressed in Equation (5). Once this concatenated vector is obtained, it is subsequently input to the action predictor to determine the action $a_{t}$.
(4)$$\begin{array}{cc} \; & a_{t}=\mathrm{Action-Predictor}\left(\left[X_{t}⊕v_{t}\right]\right) \end{array}$$
(5)$$\begin{array}{cc} \; & a_{t}=\mathrm{Action-Predictor}\left(\left[X_{t}⊕v_{t}⊕m_{t}\right]\right) \end{array}$$

## 4. Experiments

In this section, we present extensive experiments on the outdoor VLN Touchdown dataset [6] to evaluate whether our OAlM can improve the performance of models on the outdoor VLN task.

### 4.1. Implementation Details

The proposed framework and the baselines are implemented in PyTorch [32]. We used ResNet [31] to extract panorama features. For the landmark detector, we used OCR [33] and object-detection technology to recognize landmarks from the panoramas. More specifically, we used EasyOCR (https://github.com/JaidedAI/EasyOCR, accessed on 12 October 2022) to recognize the text on signboards, and we used Mask R-CNN [34] (which is based on ResNet50 [31] pretrained with the ImageNet [35] and then fine-tuned on MSCOCO [36] dataset) to detect objects from the panoramas. The landmark detector outputs results with the confidence score of each Intersection over Union (IoU), and we separated out the results scoring over 0.7, 0.8, and 0.9 to conduct experiments at each threshold value. Additionally, we extracted the ordered landmarks in the instructions (e.g., “Starbucks, MacDonald’s”) with GPT-3 [28]. Then, the extracted landmarks were lower-cased into byte-pair encodings [37] with a vocabulary of 2000 tokens and embedded with a size of 256. We then calculated the cosine similarity of these embeddings.

We used the training settings in the original papers for each model. For GA and RCONCAT, we selected the model with the best shortest-path distance (SPD) performance on the development set after 100 epochs. For ORAR, we selected the model with the best SPD performance on the development set after 150 epochs. Additionally, since the experimental results for the original ORAR code were found to be unstable, we calculated the average of the results from three experiments for evaluation.

### 4.2. Dataset

**Touchdown** [6] is a dataset that uses urban scenarios to create a large navigation environment based on Google Street View (https://developers.google.com/maps/documentation/streetview/overview, accessed on 10 September 2022). There are 29,641 nodes and 61,319 undirected edges to simulate New York City’s environment. Each node includes a 360-degree RGB panoramic image. The dataset contains 9326 trajectories for the navigation task, and each trajectory is paired with human-written navigation instructions according to the panoramas. There were 6525 samples in the training set, 1391 samples in the development set, and 1409 samples in the test set.

### 4.3. Metrics

The following metrics were used to evaluate VLN performance:(1)Task Completion (TC): the accuracy of navigating to the correct location. The correct location is defined as the exact goal panorama or one of its neighboring panoramas.(2)Shortest-Path Distance (SPD) [6]: the mean distance between the agent’s final position and the goal position in the environment graph.(3)Success Weighted by Edit Distance (SED): the normalized Levenshtein edit distance [38] between the predicted path and the ground-truth path, with points only awarded for successful paths.(4)Coverage Weighted by Length Score (CLS) [39]: a measurement of the fidelity of the agent’s path with respect to the ground-truth path.(5)Normalized Dynamic Time Warping (nDTW) [40]: the minimized cumulative distance between the predicted path and the ground-truth path.(6)Success Weighted Dynamic Time Warping (SDTW): the nDTW value where the summation is only of successful navigation.

### 4.4. Results

In this section, we report the outdoor VLN performance and the quality of object-level alignment to validate the effectiveness of our proposed OAlM. We compare the results with and without OAlM and discuss the influence of object-level alignment in outdoor VLN.

We compare the performance of our approach with the baseline GA [7], RCONCAT [6], and ORAR [10] models, since these models are not treated as specific object features on VLN.

#### 4.4.1. Quantitative Results

**OAlM with OCR:**Table 1 summarizes the results of the navigation experiments on the Touchdown dataset with and without OAlM (and using OCR as landmark detector). According to Table 1, the majority of the OAlM model’s results in each metric have outperformed the baseline ORAR. Moreover, OAlM, with an OCR score of 0.7, gave the best results in each metric. Of these, OAlM resulted in a 1% improvement in TC and showed significant improvements in the path alignment metrics (CLS, sDTW), demonstrating the effectiveness of OAlM in instruction following and goal achievement. The results of the RCONCAT and GA models show that our proposed OAlM outperformed the baseline in each metric and in each setting. Our OAlM brings about a 3% improvement in both goal-oriented metrics (TC, SED), which demonstrates that our OAlM stops better than the baselines.

**OAlM with object detection:**Table 2 summarizes the results of navigation with and without OAlM (using Mask R-CNN as a landmark detector). The top of Table 2 illustrates that most metrics were better than baseline when using the landmark detected by Mask R-CNN on the ORAR model. For RCONCAT, although TC, SED, and CLS were below the baseline in experiments utilizing 0.8 score object detection as environment landmarks. Experiment using object-detection scores of 0.9 shows a significant improvement over baseline. Additionally, for the GA model, the SPD was lower than the baseline in all experiment settings. However, other metrics are outperformed on each experiment setting. Furthermore, when given the landmarks of Mask R-CNN with 0.8 score, OAlM brings a 3% improvement in TC and SED.

#### 4.4.2. Qualitative Results

**Visualization of trajectories:** We provide visualizations of four qualitative examples to illustrate further how our OAlM learns to stop better and turn at the correct place in Figure 3. As seen in Figure 3, when navigating, the baseline tends to ignore landmarks that represent significant turning points or stopping places in instructions. Specifically, in Figure 3a,b, the baseline agent fails to understand the instructions and its current environment, causing it to stop at the wrong place. In Figure 3c, the baseline agent misses a turn and ignores the following instructions that describe the stopping place. Finally, in Figure 3d, the baseline agent turns at the wrong location and cannot understand the whole sentence, then ignores the following description of the observed environment. In contrast to the baseline, our agent stops and turns in the right places.

On the other hand, we provide some failed navigation examples in Figure 4. When we analyzed the failure cases, we found that in many instances, our model was just one step away from the goal, as seen in Figure 4a. These instances accounted for 33% of the failure cases, indicating that even in the majority of the failed scenarios, our model was remarkably close to a successful navigation path. In contrast, the failure cases for the baseline models were unevenly distributed, which we believe demonstrates an advantage of our approach. However, our model does struggle in certain complex situations. For instance, when there were two parallel paths in the environment, the agent chose the incorrect one, as shown in Figure 4b. Some cases of failure, such as Figure 4c, have difficult statements in the instructions that make the agent confused.

**OAlM is an object-focused model:** To further prove that our proposed OAlM is more object-focused than previous methods, we ran masking experiments similar to ORAR [10] and DiagnoseVLN [41]. Figure 5 shows the changes in task completion rates when masked object tokens are used. The model with the addition of OAlM shows a greater change in the task completion rate, indicating that OAlM indeed enables the model to learn objects from the instructions. However, In Figure 5a, an intriguing result can be observed at a certain data point, where the impact on our model upon masking 20% of the object tokens appears to be minimal. One possible explanation for this might be the robustness of RCOANCAT in using panorama features and instruction features, making it less dependent on specific object tokens. In this particular case, the object detection’s proficiency in extracting critical visual information might have compensated for the effects of object token masking in the instructions. Moreover, when we mask off the object token, the effect on the navigation effect is not very significant compared to ORAR (Figure 5b).

**OAlM can stop in the correct position:** As mentioned earlier, previous outdoor VLN methods tended to be off on the prediction of stop locations. In the comparison of the stop location predicted by the baselines and our OAlM, as summarized in Table 3, our OAlM has far fewer stop-location errors from failed predicted paths.

### 4.5. Discussion

The results presented above indicate that the navigation ability of agents can be improved if objects with a certain level of accuracy are used for object alignment. Although some of the experimental results were below baseline, these results were generally overly dependent on the accuracy of existing OCR and object detection. Before the experiments, we expected that the results of OAlM with Mask R-CNN would be better than OAlM with OCR, since the Touchdown dataset describes landmarks with unspecified nouns, which are similar to the results of Mask R-CNN. However, a comparison of Table 1 and Table 2 shows that OAlM with OCR actually performs better. This is because the Mask R-CNN used in this paper has only 81 classes, among which the number of classes that can be recognized from the panorama is quite limited.

In addition, we have investigated the performance of navigation when using both objects detected by OCR and object detection, and Table 4 lists the experimental results of using both OCR and object-detection results for OAlM. However, using both results did not further improve the navigational performance and irregular. Although not shown in Table 1 and Table 2, the results of the experiment using the lower threshold (lower than 0.7) landmarks were actually lower than baseline. On the other hand, Table 4 shows that when both lower scores of landmarks identified by OCR and object detection are used for OAlM, navigation is outperforming. Comparing Table 1, Table 2, and Table 4, besides the SPD in GA model, the results using only either OCR or object detection yield the best results. This is because not all objects detected by OCR and object detection are correct, and the match score is likely to be noise. Moreover, our OAlM is a simple model that cannot handle such complex combinations. In the future, we plan to develop a model that can use multiple object information in a balanced manner.

Based on the quantitative and qualitative results, our proposed OAlM solves the problem of ignoring objects that existed in previous methods. The OAlM method also improves the navigation capability of the agent by focusing on the objects in the instructions and panoramic images.

## 5. Conclusions

We introduced a novel object-level alignment module for outdoor VLN and showed that explicit landmark-level alignment benefits VLN in an outdoor environment, which is considerably more complex and diverse than indoor environments. Our results demonstrated the effectiveness of using OAlM with three baseline models. Experiments illustrated that object-level alignment can improve outdoor VLN performance, indicating that objects from natural language instructions and the environment are important clues in outdoor VLN. Our proposed OAlM has the advantage of being more interpretable for humans and more accurate in navigation, both in determining where to turn and in stopping.

**Limitation:** Our OAlM extracted and detected landmarks from instructions and the corresponding environment. A shortcoming in our work is that our landmark detector did not use OCR and object detection was retrained for outdoor environments. However, from our current experiments, we have found that focusing on objects is useful for outdoor navigation. A further shortcoming is that our model’s use of both OCR and object-detection features will result in reduced performance, while separately they both contribute positively to the navigation performance. This may be due to the lack of a special structure designed to handle these two features simultaneously and effectively.

In the future, we plan to focus on investigating how to improve our model’s ability to process both OCR and object-detection inputs effectively simultaneously. We believe that existing OCR and object-detection techniques should be applied to outdoor environments in order to make more use of objects in the environment.

## Figures and Tables

**Figure 1 sensors-23-06028-f001:**
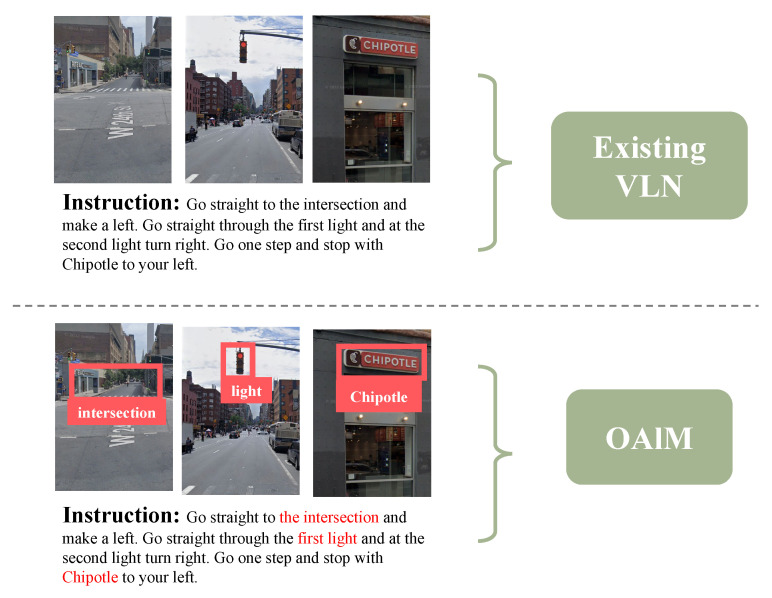
Comparison of existing outdoor vision-and-language navigation models with simple feature concatenation (**Top**) and our object-alignment module focused on landmarks for navigation (**Bottom**).

**Figure 2 sensors-23-06028-f002:**
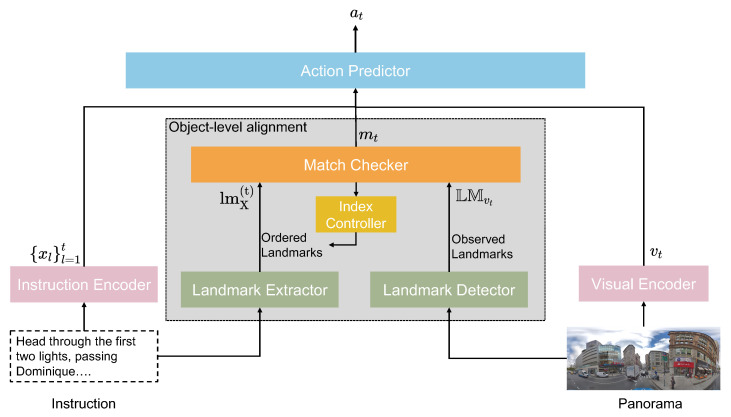
Illustration of the proposed **O**bject-level **Al**ignment **M**odule (OAlM), using instruction feature, visual features, and the match score of landmarks to predict action at each time.

**Figure 3 sensors-23-06028-f003:**
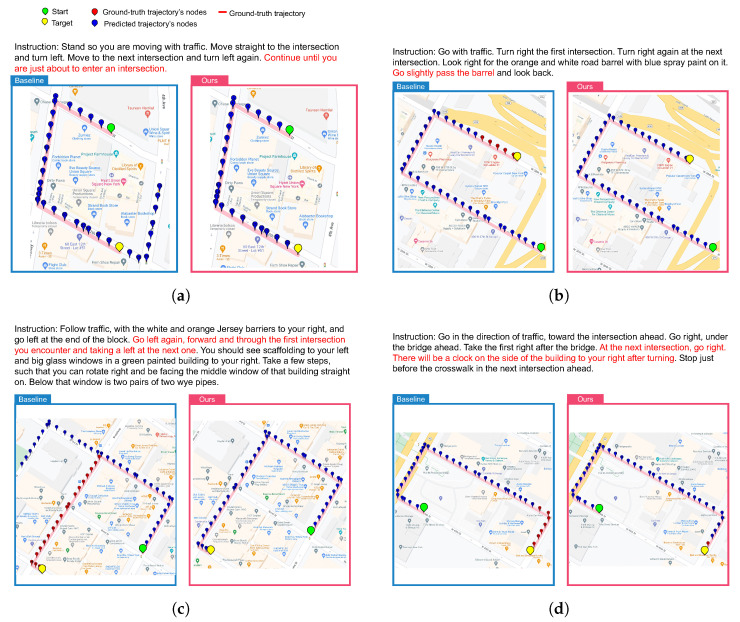
Visualizations of baseline and OAlM models. Two examples from the test set are shown where our approach is successful, but the baseline model stops too late or turns at the wrong place. Left: trajectory generated by baseline. Right: trajectory generated by our OAlM. (**a**) A case where the baseline model stops too late. (**b**) A case where the baseline model stops too early. (**c**) A case where the baseline model misses a turn. (**d**) A case where the baseline model turns at the wrong place.

**Figure 4 sensors-23-06028-f004:**
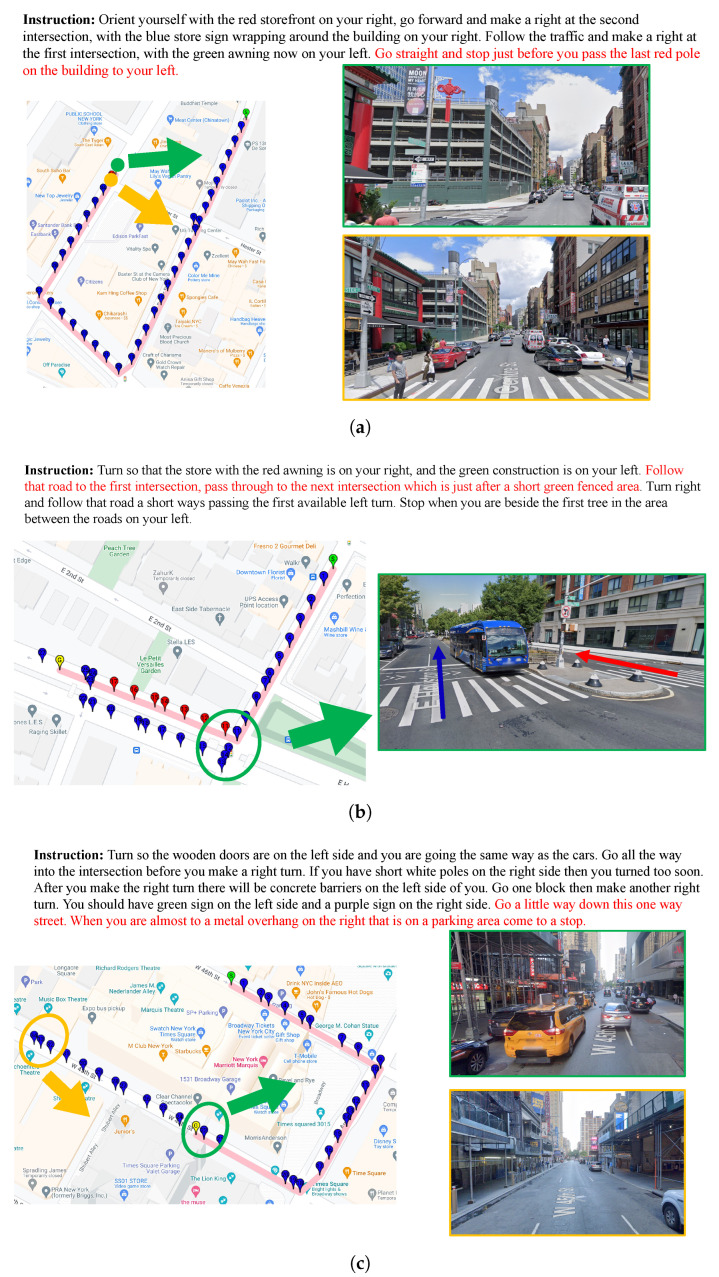
Visualization of the failed navigation examples; the top part of each figure is the instruction, the left part represents the trajectory generated by our OAlM, and the right part represents the scene observed by the agent during partial real navigation. The green or orange circled parts in each figure on the left correspond to the specific scenario on the right, which is also marked in green or orange. (**a**) A case where the OAlM model stops early and is just one step away from the goal. (**b**) A case of navigation failure due to the complex road conditions. The red arrow in the right figure shows the correct road. The blue arrow shows the model-predicted road. (**c**) A case where the OAlM model’s inaccuracy causes navigation failures.

**Figure 5 sensors-23-06028-f005:**
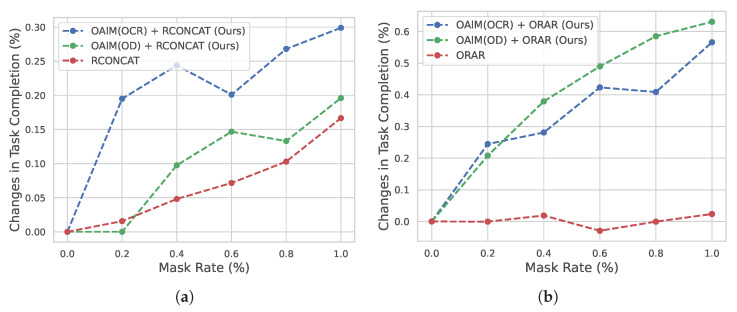
Navigation performance when masking out object tokens from instructions. (**a**) Change in TC for RCONCAT. (**b**) Change in TC for ORAR.

**Table 1 sensors-23-06028-t001:** Navigation results when adding OCR to OAlM. The blue numbers in the table indicate results that were worse than the baseline, while the red numbers indicate results that were better than the baseline. Bold numbers indicate the highest score in the column (per model).

Experiment	TC↑	SPD↓	SED↑	CLS↑	nDTW↑	sDTW↑
**ORAR** (baseline)	24.23	17.30	23.70	56.87	37.20	22.87
**OAlM (+OCR 0.70)**	**25.86**	**16.94**	**25.27**	**58.21**	**38.23**	**24.29**
**OAlM (+OCR 0.80) **	24.51	17.39	23.92	56.53	36.37	23.06
**OAlM (+OCR 0.90) **	25.84	17.45	25.17	56.37	37.45	24.20
**RCONCAT (baseline)**	8.94	22.48	8.55	43.23	18.20	7.98
**OAlM (+OCR 0.70) **	**11.64**	21.15	**11.26**	44.11	20.95	**10.74**
**OAlM (+OCR 0.80) **	9.94	21.47	9.67	**44.28**	20.34	9.21
**OAlM (+OCR 0.90) **	11.07	**20.99**	10.41	44.19	**21.10**	9.94
**GA** (baseline)	9.87	20.34	9.42	47.77	21.51	8.92
**OAlM (+OCR 0.70) **	10.36	20.58	9.91	48.01	22.55	9.24
**OAlM (+OCR 0.80) **	**11.43**	20.20	**11.11**	**49.03**	**23.84**	**10.51**
**OAlM (+OCR 0.90) **	10.93	**19.62**	10.63	47.85	22.68	10.09

**Table 2 sensors-23-06028-t002:** Navigation results adding object-detection results to OAlM. The blue numbers in the table indicate results that were worse than the baseline, while the red numbers indicate results that were better than the baseline. Bold numbers indicate the highest score in the column (per model).

Experiment	TC↑	SPD↓	SED↑	CLS↑	nDTW↑	sDTW↑
**ORAR (baseline)**	24.23	17.30	23.70	56.87	37.20	22.87
**OAlM (+OD 0.70)**	24.70	16.76	24.10	57.66	36.99	23.04
**OAlM (+OD 0.80)**	**24.94**	17.08	**24.32**	56.91	37.06	**23.39**
**OAlM (+OD 0.90)**	24.72	**16.52**	24.12	**58.33**	**37.63**	23.20
**RCONCAT (baseline)**	8.94	22.48	8.55	43.23	18.20	7.98
**OAlM (+OD 0.70)**	9.65	21.84	9.31	43.40	20.24	8.83
**OAlM (+OD 0.80)**	8.66	21.45	8.43	42.87	20.52	8.11
**OAlM (+OD 0.90)**	**10.15**	**20.56**	**9.77**	**45.99**	**21.22**	**9.23**
**GA (baseline)**	9.87	**20.34**	9.42	47.77	21.51	8.92
**OAlM (+OD 0.70) **	11.57	20.39	11.23	47.89	22.61	10.60
**OAlM (+OD 0.80)**	**13.06**	20.36	**12.56**	**48.49**	**23.36**	**11.86**
**OAlM (+OD 0.90)**	10.86	20.79	10.57	48.17	22.98	10.06

**Table 3 sensors-23-06028-t003:** Percentage of stop-location errors of failed predicted paths.

	Baselines	OAlM
**ORAR**	32.76	4.38
**RCONCAT**	15.34	7.87
**GA**	23.23	5.84

**Table 4 sensors-23-06028-t004:** Navigation results adding both OCR and object-detection results to OAlM. The blue numbers in the table indicate results that were worse than the baseline, while the red numbers indicate results that were better than the baseline.

Experiment	TC↑	SPD↓	SED↑	CLS↑	nDTW↑	sDTW↑
**ORAR (baseline)**	24.23	17.30	23.70	56.87	37.20	22.87
**+OCR&OD 0.40**	**24.96**	**16.53**	**24.48**	**58.65**	**38.06**	**23.48**
**+OCR&OD 0.50**	24.81	16.85	24.10	57.08	36.90	23.19
**+OCR&OD 0.60**	22.73	18.20	22.23	55.02	35.01	21.41
**+OCR&OD 0.70**	22.92	16.90	22.87	57.41	36.37	21.43
**+OCR&OD 0.80**	23.35	17.10	22.82	57.14	36.50	21.99
**+OCR&OD 0.90**	22.27	18.47	21.73	53.57	34.34	20.85
**RCONCAT (baseline)**	8.94	22.48	8.55	43.23	18.20	7.98
**+OCR&OD 0.40**	**10.15**	21.45	**9.58**	43.63	20.05	**8.90**
**+OCR&OD 0.50**	9.44	21.51	9.03	44.08	**20.25**	8.49
**+OCR&OD 0.60**	8.52	21.78	8.28	42.30	18.82	7.91
**+OCR&OD 0.70**	9.01	21.69	8.55	**44.46**	19.05	7.73
**+OCR&OD 0.80**	9.65	21.45	9.20	44.43	20.24	8.56
**+OCR&OD 0.90**	8.87	**21.02**	8.49	44.35	20.00	7.93
**GA (baseline)**	9.87	20.34	9.42	47.77	21.51	8.92
**+OCR&OD 0.40**	11.50	19.82	11.19	**49.19**	**23.76**	10.57
**+OCR&OD 0.50**	10.01	19.70	9.73	48.33	22.17	9.22
**+OCR&OD 0.60**	**11.71**	**19.17**	**11.28**	49.05	23.67	**10.69**
**+OCR&OD 0.70**	7.67	23.23	7.34	42.95	16.32	6.74
**+OCR&OD 0.80**	9.79	21.47	9.44	46.44	20.60	8.80
**+OCR&OD 0.90**	10.43	20.97	10.17	46.19	20.63	9.70

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
