# Peer review of "Outdoor Vision-and-Language Navigation Needs Object-Level Alignment"

_sensors, 2023, doi:10.3390/s23136028_

Round 1

Reviewer 1 Report

The authors present an interesting paper in wich they propose a novel approach to determine outdor localization and navigation method based on Vision-and-Language Navigation. However, it is advisable that they implement some improvements, namely:

1. - Bibliographic reference are out of order. They are not in alphabetical or numerical order. References [6, 15, 20, 23, 24, 25, 26] appear in bibliographic list at end of document, without, however, being referred across the text.

In this sense, as these references are important, the authors should rewrite some parts of the docuent, in order to incorporate them in a coherent way.

2. - The inference/inclusion of the OAIM methos should be better exposed/explained, is, better substantiated.

3. - The legend of the Tables (1, 2, 3), should be checked, as it seems to have some discrepancies in relation to the numerical part presented in the table. When authors refer to "higher" do they mean "better"? It should be the labels "better" and "worse" instead of "lower" or "higher", because when the trend is decreasing, the numerical part and legend are not consistent with each other.

4. - when showing fig. 4, the authors should substantiated or indicate and identify cleary the reasons why, during a period the presented method is inferior to the two reference methods, after which it remains in an intermediate situation between the other two.

5. - In the conclusions, the authors state "our proposed OAIN has the advantage ...", without, however, making any comments to explain the following anomalous situatio: the proposed method with OD or OCR (separatly) is better, generically, than the reference methods, but in the case of considering OD & OCR (0.7) it present worse results than the reference methods. There is no reference to this subject in the text!

Finally, congratulations on your work!

Author Response

We greatly appreciate your comments and respond to all comments in the attached PDF file. 

Reviewer 2 Report

In this paper a vision-and-language navigation system is presented. It is based on images, a textual instruction for a navigation of an agent. Tha authors propse to enhance the performance of the agent with reaching destination points by extracting the landmarks from visual data, and align them with mentioned in text instruction, and set as an additional input for action predictor. Such approach helps the agent to navigate more reliably. Authors compare the existing solutions with the proposed approach, and enhanced metrics. the relted works are clearly presented, as well as proposed system improvements.

I have the following questions and notes:

1. In section with discussion it is mentioned, that optical caracter recognition system for street signs (and other text items) works better that approach, based on general object detection (like Mask-R CNN). However, in figure 1 the bottom illustration suggests that agent relies more on objects, that street signs. 

2. Does the metrics improvements comes from the algorithm, or from the nature of test set, where probably there are numerous examples with text signs?

3. The abbreviations OAAM and ROAA should be explained. Actually, there are a lot of other abbreviations of existing approaches.

4. Line 294 probably should contain some missed text

5. The examples in Figure 3 contain only the cases when the proposed approach is better. It would be interesting to see in which cases it failed, and some assumptions why it happens.

Author Response

(The authors gave the same response as above.)

Round 2

Reviewer 1 Report

The authors implemented, in a generic way, all the sugestions and recommendations made, so the paper meets the conditions for publication.

Congratulation on your work.